# The Predictive Utility of Valuing the Future for Smoking Cessation: Findings from the ITC 4 Country Surveys

**DOI:** 10.3390/ijerph19020631

**Published:** 2022-01-06

**Authors:** Ron Borland, Michael Le Grande, Bryan W. Heckman, Geoffrey T. Fong, Warren K. Bickel, Jeff S. Stein, Katherine A. East, Peter A. Hall, Kenneth Michael Cummings

**Affiliations:** 1School of Psychological Sciences, University of Melbourne, Parkville 3010, Australia; mlegrande@unimelb.edu.au; 2Department of Psychiatry & Behavioral Sciences, Medical University of South Carolina, Charleston, SC 29425, USA; bheckman@mmc.edu (B.W.H.); cummingk@musc.edu (K.M.C.); 3Center for the Study of Social Determinants of Health, Meharry Medical College, Nashville, TN 37208, USA; 4Department of Psychology, University of Waterloo, Waterloo, ON N2L 3G1, Canada; geoffrey.fong@uwaterloo.ca; 5School of Public Health Sciences, University of Waterloo, Waterloo, ON N2L 3G1, Canada; katherine.east@kcl.ac.uk (K.A.E.); pahall@uwaterloo.ca (P.A.H.); 6Ontario Institute for Cancer Research, Toronto, ON N2L 3G1, Canada; 7Fralin Biomedical Research Institute at Virginia Tech Carilion, Roanoke, VA 24016, USA; wkbickel@vtc.vt.edu (W.K.B.); jstein1@vtc.vt.edu (J.S.S.); 8Department of Addictions, Institute of Psychiatry, Psychology and Neuroscience, King’s College London, London SE5 8AF, UK

**Keywords:** delay discounting, time perspective, smoking cessation, longitudinal study, financial stress

## Abstract

Background: Delay discounting (DD) and time perspective (TP) are conceptually related constructs that are theorized as important determinants of the pursuit of future outcomes over present inclinations. This study explores their predictive relationships for smoking cessation. Methods: 5006 daily smokers at a baseline wave provided 6710 paired observations of quitting activity between two waves. Data are from the International Tobacco Control (ITC) smoking and vaping surveys with samples from the USA, Canada, England, and Australia, across three waves conducted in 2016, 2018 and 2020. Smokers were assessed for TP and DD, plus smoking-specific predictors at one wave of cessation outcomes defined as either making a quit attempt and/or success among those who tried to quit which was ascertained at the subsequent survey wave. Results: TP and DD were essentially uncorrelated. TP predicted making quit attempts, both on its own and controlling for other potential predictors but was negatively associated with quit success. By contrast, DD was not related to making quit attempts, but high DD predicted relapse. The presence of financial stress at baseline resulted in some moderation of effects. Conclusions: Understanding the mechanisms of action of TP and DD can advance our understanding of, and ability to enhance, goal-directed behavioural change. TP appears to contribute to future intention formation, but not necessarily practical thought of how to achieve goals. DD is more likely an index of capacity to effectively generate competing future possibilities in response to immediate gratification.

## 1. Introduction

Smoking cessation is a behaviour that involves weighing a choice for an immediate reward (i.e., nicotine satisfaction and/or avoid nicotine withdrawal) against some future benefit (i.e., reduced risk to health, cost savings). Although this temporal tradeoff is often recognized as a major obstacle to smoking cessation, to date, there has been less research in this domain than would be suggested by its centrality.

There are two general approaches to understanding the role of intertemporal choice and judgment. Delay reward discounting (DD) is a depreciation of the value of a reward as a function of the time that it is received [1]. As typically assessed, DD assesses the extent to which respondents discount future monetary rewards. For example, when offered USD 1000 in one year’s time, most people will accept considerably less to obtain the money today. Indeed, they will typically accept less on average than the going interest rate, meaning that to obtain the immediate reward they are prepared to lose money in the longer term. As this preference for immediate rewards increases, the discounting rate becomes steeper (i.e., greater discounting), which indicates higher levels of impulsivity [2] an underlying dispositional characteristic manifest as valuing the present over the future. More generally, DD for money is theorized to be positively related to discounting other aspects of the future, in this case the consequences of smoking [3]. As a result, we would expect less success in quitting smoking in those with higher DD rates for cash because it reflects an over-valuing of immediate outcomes over longer term consequences [3,4,5]. This thinking has led to DD being used as a behavioral marker of addiction [6].

Cigarette smokers discount the future to a substantially greater extent than non-smokers [3,7]. DD has been associated with a range of smoking cessation outcomes from intention to quit [4] to reduced likelihood of relapse [8]. DD is positively correlated with dependence level as indexed by the Fagerstrom test of nicotine dependence (FTND) [9] and with cigarettes smoked per day [10,11]. Ex-smokers also report lower DD than current smokers [3,12]. These findings suggest a direct influence on cognitive processes. In one study, DD was a stronger predictor of relapse than conventional measures of dependence [8], suggesting a more direct effect on relapse prevention.

Conceptually related to DD is the concept of time perspective [13,14]. Hall and Fong theorize that the extent to which people consider the future in decision making is a key underlying factor behind health risk behaviours such as smoking. Time perspective (TP) is assessed by explicitly asking respondents about extent to which their thinking focuses on implications of current actions for the future, with higher scores being indicative of having a future orientation. TP is the conceptual opposite of DD, if DD is considered as a measure of devaluing the future, so it should be closely, but inversely correlated with DD. However, if DD is considered to be more as a measure of impulsiveness, there is less reason to expect them to be correlated. TP also differs from DD in that it is a belief about the disposition, in this case about having a future orientation, rather than the more direct measure of valuing the future in DD, i.e., based on choosing less reward when offered immediately as compared with when only available in the future. TP, the belief, is theorized to impact on intentions, but not on behaviour at least directly [15]. Future oriented TP has been shown to be predictive of a variety of health protective behaviors [16,17,18] and non-smoking status [19]; making quit attempts [20,21] and quit success with history of engagement in quit attempts mediating this relationship [21], but not studied when tested only among those trying.

The distinction between beliefs and dispositions mirrors the distinction in social psychology between explicit and implicit attitudes [22]. Implicit attitudes focus on the disposition to act in particular ways and are typically assessed independent of self-report, while explicit beliefs and attitudes are those reported and thus are designed to reflect thinking about the issue (i.e., what they say they believe). Studies of explicit and implicit measures of essentially the same construct show varying levels of association from high to non-existent, and a discrepancy between the two has been linked to difficulty in changing related behaviours [23,24].

One way to analyze possible mechanisms for differences between beliefs and related dispositions comes from CEOS theory. CEOS is an acronym for context, executive, and operational systems [25,26]. CEOS theory focuses on the interaction between conscious and non-conscious influences on behaviour and is aimed at explaining and exploring the limits of executive control and thus self-management of behaviour. There is substantial evidence for a dual process approach [27,28], including of specific neural underpinnings [29].

CEOS theory treats dispositional measures as indicators of how the person behaves or will respond to contextual stimuli (objects, people, activities) that are measured independent of conscious beliefs. It locates dispositions within its operational system and explains their mode of influence in terms of the affective forces they generate towards particular action patterns. Operational forces can be inferred from actual behaviour or, in the absence of spontaneous behaviour, experienced feelings and urges to act (sometimes described as “what we want to do”). By contrast, CEOS theory treats explicit attitudes as reports on evaluative beliefs about the conceptual desirability or otherwise of the activity (“what we should do”). Behaviour-related beliefs can be grounded in an attempt to reconcile some or all factual information, social expectations, and relevant experiences into a coherent position, or simply be a rationalization of experienced desires. In the case of evidence-grounded beliefs, there can be a discord with experienced dispositional tendencies. CEOS defines hard-to-change behaviours as ones where there is a discord between desires based on existing dispositions and those based on evaluative beliefs (i.e., between “should” and “want”).

According to the theory, affective forces (operationally generated desire) and the tendencies towards action they produce (experienced as urges) are the primary determinants of behaviour [26] and thus beliefs can only affect behaviour by means of stimulating operational desires to act. This is performed by linking ideas to desired outcomes by creating and using stories about possibilities that evoke the relevant desires.

CEOS theory also postulates that executive beliefs only influence behaviour (smoking in this case) when they are brought to mind. Because goal seeking is about future outcomes, it follows that the capacity of the person to consider and value future events relative to current ones should affect the potency of future-oriented beliefs on behaviour. However, it is unclear whether such tendencies would be incorporated within the plans people make, or independently affect the likelihood of plans translating into action.

Most smokers have a strong emotional attachment to smoking, a conditioned response built up from smoking thousands of cigarettes (i.e., they have a strong disposition to smoke); that is, it has become a habit. The difficulty most smokers experience in trying to quit [30,31] demonstrates that smoking is also sustained by the body’s response to nicotine, a dependence producing drug [32,33]. The combination of habitual factors and the biological dependence makes smoking cessation particularly difficult which means it requires high levels of executive-generated effort by the smoker, if smoking is to be successfully resisted [32,33,34,35]. This requires a powerful story where both motivation and persistence are needed to achieve the goal of smoking cessation. Where this effort is successful it can increase the strength of affective forces for not smoking while reducing the strength of the desire to smoke—thus, making not smoking gradually easier overtime. However, where quitting is experienced negatively, including feeling it is too difficult to refrain from smoking, the smoker’s effort to resist smoking can be compromised and a competing rationalizing story can come to dominate with a focus on the losses associated with quitting and or with dissonance-reducing thoughts (i.e., thoughts that ignore or downplay information about the harms) which justify a resumption of smoking. Recent work suggests that avoidance of thinking about information is a common and surprisingly easy strategy [36].

Harder to imagine scenarios are discounted more than ones that can be more vividly imagined [37,38,39] demonstrating that the imaginability of future scenarios is important determinant of discounting. There is also evidence that measured DD can be modified by relevant experiences. In one study, the use of contingency management to reduce cigarettes smoked per day led to decreased discounting of future rewards [40], because less money is required to meet current needs, so the present value of money has declined. An alternative explanation is that the immediate rewarding effects of cigarettes tends to focus attention on the short-term reward, while abstaining from smoking focuses attention on longer term gains [41]. Unless and until smokers can imagine and anticipate the consequences of smoking, they may lack sufficient motivation to abstain. Laboratory research has shown that having cigarette-deprived smokers engage in episodic future thinking reduces the amount they smoke [39]. This mechanism, consistent with the above theorizing, can also explain why smokers who are personally affected by smoking related illness are more likely to attempt to quit [42], and to why smokers respond more to anti-advertising that is emotionally arousing than to the presentation of uncontextualized facts about the dangers of smoking [43].

In summary, because TP and DD are associated with different mechanisms [25,44,45], the above analysis suggests that they may play different roles in making and sustaining attempts to quit smoking. DD may be better conceptualized as measuring aspects of impulsivity/restraint capacity generated in part by the capacity to image futures sufficiently to complete with immediate urges than a generic capacity to value the future. In contrast, TP may be conceptualized as measuring conscious estimates of thinking about the effects of present activity on possible futures and as currently operationalized does not have a strong focus on what the future is likely to be or of how to achieve it. This distinction between DD and TP suggests that there may be only a moderate correlation between the two constructs, owing to their close conceptual links related to future thinking, but diverging in the specific aspects of their common origins. We would expect TP to predict interest in, but not concerted action towards, behaviours with long term payoffs, unless TP is also tapping impulsivity, in which case any relationship should disappear once DD was controlled for as DD more directly measures aspects of impulsivity. By contrast, for making quit attempts the reverse might be the case with the TP belief likely to mediate any relationship between trying to quit and DD as trying is primarily driven by executive processes.

There are several potential moderators of the associations between DD and quitting and to the extent to which they are related, may also apply to TP. Measures of DD involving money are likely to vary as a function of the person’s current economic circumstances. The predictive power of DD is also theorized to be on top of, and potentially interactive with the difficulty of the behaviour change task [9,10,11]. Finally, there is evidence of age -related effects on quitting outcomes [46] and both DD and TP arguably have different connotations as a function of age. We, therefore, specifically looked for interactive effects of baseline financial stress, dependence, motivation, self-efficacy, and age.

Thus, the three aims and associated hypotheses of this paper are to explore:The association between TP and DD: we predict that TP and DD will be moderately negatively correlated; that is, high DD being associated with lower levels of TP.The predictive effects of TP and DD for making a quit attempt and for sustained smoking abstinence. We predict, based on past findings, that TP will predict making quit attempt, while DD will predict success among those who try. However, we are uncertain as to whether these two measures will add explanatory power to prediction in isolation or when major known determinants of each outcome are controlled for.Whether the relationship of TP and DD to quitting and sustaining abstinence from cigarettes is moderated by sociodemographic factors, financial stress, measures of dependence, and motivation. We make no specific predictions.

## 2. Materials and Methods

### 2.1. Data Source and Study Population

The current study combined and analysed three waves of the International Tobacco Control (ITC) Four Country Smoking and Vaping Survey (Wave 1, 2016; Wave 2, 2018; Wave 3, 2020). Full descriptions of the ITC Project conceptual framework and methods have been published elsewhere [47,48]. Briefly, the cohorts come from four countries: England, US, Canada, and Australia and the data were collected using a mix of telephone interviews and web surveys. For this analysis, the analytic sample consisted of those smokers (n = 3511 W1, n = 1495 W2 replenishments; total n = 5006) who provided data for at least one wave-to-wave transition (n = 6710 observations; 3511 observations from W1 to W2 and 3199 observations from W2 to W3). Respondents were eligible for inclusion for the analyses in the current study if they were current daily smokers aged 18 and over who provided heaviness of smoking (HSI) data at baseline. In keeping with our previous work, we chose to exclude those who were baseline daily vapers as vaping levels are likely to affect the dependence measures and DD may differ between vapers and non-vapers, independent of the association between cigarette smoking and non-smoking [12].

### 2.2. Measurements

Demographic covariates. Demographics included age at recruitment (18–24, 25–39, 40–54, 55–max), gender, and country (England, US, Canada, Australia). Highest level of education attained was coded into low, medium, and high. Financial stress was assessed by the question “In the last 30 days, because of a shortage of money, were you unable to pay any important bills on time, such as electricity, telephone or rent bills?” We choose to use this variable over household income since it has less missing data (1.2% compared to 5.4% for income) and this variable was associated with income but is perhaps a better measure of current economic circumstances. We also adjusted for self-imposed smoking restrictions in the home since this has been found to explain some of the predictive value of time to first cigarette [49]. All respondents at Wave 1 were asked “Which of the following best describes smoking cigarettes inside your home?” with response options ‘Smoking is allowed anywhere in your home’, ‘Smoking is NEVER allowed ANYWHERE in your home’, ‘Something in between’.

### 2.3. Dispositional Measures

Delay discounting. DD was assessed by asking respondents questions about which of two amounts of hypothetical money they would prefer, a delayed USD 1000 (the larger later option) or a reduced immediately available amount USD 500 (the smaller sooner option). An example item is “Would you rather have USD 500 now or USD 1000 in 1 day?” In England Pounds replaced dollars), but no adjustment was made for currency values. The delays ranged from 1 h to 25 years and adjusted across five contiguous trials depending on responses to earlier questions [50]. Although all reward choices were hypothetical, this task has convergent validity with choices that are actualized [51,52]. Prior to analyses reported below, temporal discounting values (k) were approximately normalized using the natural-log transformation. However, to aid interpretation we report means with a constant (+10) added to make all data positive. We also divided this value into tertiles (low, medium, high) for the purpose of describing the bivariate relationships of DD with covariates that was approximately compatible with the three levels of time perspective as reported in Table A1.

Time perspective. TP was assessed using a single item from the time perspective questionnaire (TPQ) [14]. Respondents were asked to rate on a five-point Likert scale how much they agree or disagree with the statement “You spend a lot of time thinking about how what you do today will affect your life in the future”; with agreement reflecting greater future-oriented time perspective. The TPQ scale has demonstrated adequate reliability and validity, and the item drawn from it has a strong item-total correlation with the full scale [13]. Because of low numbers in the extreme categories (strongly agree and strongly disagree), both less than 8%, the scale was recoded into a three-category scale—disagree, neutral, agree.

### 2.4. Motivational Measures

Intention to quit: Intention to quit smoking was measured using the item “Are you planning to quit smoking…” with the response options “Not planning to quit”, “Sometime in the future, beyond 6 months”, “Within the next 6 months”, ”Within the next month” and “don’t know”. A binary version of this measure was constructed comparing “not planning to quit” with all other options.

Wanting to quit: Wanting to quit, a measure of explicit motivation was measured by the item “How much do you want to quit smoking?” with the response options “not at all”, “a little”, “somewhat”, and “a lot” and “don’t know”. A binary version of this measure was also constructed with “not at all” compared to all other responses. Since there was only a moderate correlation between wanting to quit and intention to quit in this analytic sample (r = 0.45) this measure can be considered independent from intention to quit.

Quitting self-efficacy: Self-efficacy was measured using the item “If you decided to give up smoking completely in the next 6 months, how sure are you that you would succeed?” and the 5-point response scale was “Not at all sure/Slightly sure/Moderately sure/Very sure/Extremely sure”. “Don’t know responses were coded as “Not at all sure”.

### 2.5. Measures of Dependence

Heaviness of smoking index (HSI) (continuous): To avoid restrictions of the imposed categorical classification of continuous data [53], we used the methods described in Borland et al. [54] to transform the two components of HSI (cigarettes per day using square root transformation and time to first cigarette in minutes using the natural log) into a normally distributed continuous variable (HSI continuous). For our analytic sample this approach was preferable to using the standard categorical version of the HSI, since it avoided the problem of low or zero cell numbers in sub-group analysis, particularly in the extreme HSI categories.

Perceived addiction to smoking: All regular smokers were asked “Do you consider yourself addicted to cigarettes?” with the response options: ‘not at all’, ‘yes, somewhat addicted’, ‘yes, very addicted’ and ‘don’t know’. This measure was adapted from an earlier five-category measure in the cigarette dependence scale [55]. For prediction of smoking abstinence, the ‘not at all’ category for perceived addiction was too small for meaningful analysis (n = 25) and were combined with the ‘somewhat’ category, thus comparing ‘very addicted’ versus all other responses.

Strength of urges to smoke in past 24 h: At baseline all regular smokers were asked “In general, how strong have urges to smoke been in the last 24 h?” with responses forming a six-point Likert scale (0—I have not felt the urge to smoke in the last 24 h; 1—slight; 2—moderate; 3—strong; 4—very strong; 5—extremely strong and ‘don’t know’). This variable was treated as an ordinal measure with don’t know (less than one percent of responses) treated as missing.

### 2.6. Outcomes

The two outcomes assessed in this study were:Reporting a quit attempt of any duration between Waves, and;Reporting successful abstinence from smoking for one month or longer, and separately for 6 months or longer, independent of current status, between baseline and follow-up, among those smokers who had made a quit attempt. Those quit for less than the criterion were treated as quit failures, and those still quit at follow-up but for less than the criterion were excluded from analysis. At least one month abstinence was chosen since short unsuccessful quit attempts may not be recalled successfully [56] and 6 months was also included to assess stability of estimates and to relate to the duration of abstinence commonly used to define successful cessation in clinical trials.

### 2.7. Data Analysis

Three main sets of analyses were conducted.

Bivariate analyses evaluated the association of time perspective and delay discounting (tertiles) with baseline sociodemographic, dependence measures and motivational variables. We also evaluated the association of all variables with the follow-up outcomes (quit attempts, one month and ≥six months abstinence). Pearson Chi-square was used to evaluate categorical variables and independent t-tests were used for continuous measures.Separate logistic regression models for estimating probability of follow-up quit attempts and ≥1 month smoking abstinence were estimated using generalised estimating equations (GEE) with an unstructured correlation matrix, robust standard errors and the binary logit link [57]. The primary predictor variable for all models were TP and DD. All models adjusted for demographics measures (age group, gender, country, ethnicity, education level and perceived financial stress), dependence measures (HSI, strength of urges to quit and perceived addiction to smoking) and motivational measures (planning to quit, wanting to quit and perceived quit-efficacy). We additionally adjusted for self-imposed restrictions on smoking in the home and non-daily vaping frequency.We tested for interactions of the time perspective and delay discounting measures by age group (split at 40) [46], country, gender, education, ethnicity, financial stress and HSI for both quit attempts and smoking abstinence).

All confidence intervals were computed at the 95% confidence level. Analyses were conducted using Stata V16.1 (StataCorp, College Station, TX, USA).

## 3. Results

The bivariate associations between time perspective, delay discounting and study covariates are presented in Appendix A. While DD and TP were associated, the magnitude of the relationship was small and positive (r = 0.04, *p* = 0.001), unlike the hypothesized negative relationship (i.e., those who think more about the future did not discount it less). Higher TP was associated with younger age (*p* < 0.001), higher education level achieved (*p* < 0.001), reporting financial stress (*p* < 0.001), both planning and wanting to quit (*p* < 0.001), higher quit efficacy (*p* < 0.001), higher perceived addiction to smoking (*p* = 0.031) and higher mean strength of urges to smoke (*p* < 0.001), and lower mean HSI (*p* = 0.009). Higher DD was associated with younger age (*p* < 0.001), female sex (*p* < 0.001), lower education level achieved (*p* < 0.001), reporting financial stress (*p* < 0.001), not wanting to quit (*p* = 0.026), higher strength of urges to smoke (*p* < 0.001), and higher HSI (*p* < 0.001).

There were also country differences (all *p* < 0.001) with TP was highest in Canada and lowest in England, and DD highest in Australia and lowest in England (Table A1).

Overall, TP was weakly positively associated with plans to quit (r = 0.25, *p* < 0.001) and wanting to quit (r = 0.27, *p* < 0.001), but DD was not significantly related to either variable: plan to quit (r = 0.01, *p* = 0.343); want to quit (r = −0.00, *p* = 0.522) (Table A1).

Overall, 41.9% of the smokers (n = 2810) reported a quit attempt between baseline and follow-up. Of those smokers who made a quit attempt, 37.3% remained abstinent for at least one month and 17.8% remained abstinent for at least 6 months. Table 1 shows the bivariate associations between baseline demographic and predictor measures and reporting a quit attempt and 1 or 6 month sustained smoking abstinence at follow-up. Reporting a quit attempt was associated with TP with those who think more about the future more likely to make quit attempts than those who do not. Among those smokers who made a quit attempt, a higher TP was associated with inability to sustain at least one month or six months smoking abstinence. Mean DD did not differ significantly between those who reported and did not report quit attempts, but among those who made an attempt, lower DD was associated with successful abstinence at both one month and six months.

Table 2 shows the fully adjusted GEE models that estimate odds for making quit attempts and sustained smoking abstinence. Reported quit attempts remained significantly positively associated with TP. In contrast, maintaining at least one month smoking abstinence was no-longer significantly negatively associated with TP, however, there was a non-significant trend. There was no evidence of a linear relationship with TP for quit success. For the 6 months smoking abstinence outcome a neutral TP was associated with success matching the unadjusted association. As with the unadjusted results, DD was not associated with quit attempts. As predicted, lower DD was associated with successful smoking abstinence for both the one month and six months abstinence criteria.

Turning to the moderator analyses, individual tests for interactions in full models with all covariates (Table 2) revealed significant interactions for TP by financial stress both for 1 month abstinence (aOR = 0.72, 95% CI 0.51–0.99, *p* = 0.046) and for six months abstinence (aOR = 0.56, *p* = 0.023, 95% CI 0.34–0.92). Higher TP was associated with higher likelihood of achieving abstinence in financially distressed smokers but reduced likelihood of abstinence in the non-financially distressed. There was also a significant DD by financial stress interaction for one month abstinence (aOR = 0.88, 95% CI = 0.79–0.99, *p* = 0.031) with lower DD associated with abstinence in the non-financially distressed smokers in contrast to DD being positively associated with abstinence in smokers reporting financial distress. This interaction was not evident for six months abstinence (aOR = 1.04, 95% CI = 0.88–1.23, *p* = 0.629). No significant interaction was observed between TP and DD for one month abstinence (aOR 1.02, 95% CI = 0.97–1.08, *p* =0.403) and six months abstinence (aOR 1.00, 95% CI = 0.93–1.08, *p* =0.996). There were no other significant interactions for TP or DD with age, gender, country, education level attained, ethnicity and HSI for either quit attempts or smoking abstinence.

We also conducted sensitivity analyses for the final GEE models by inclusion of previous wave measures of NRT use, quit fatigue, perceived damage of smoking to health, and worry smoking will damage your health. For both one month and six-month smoking abstinence the inclusion of these variables did not significantly affect the aOR for DD individually or when all of these measures were added to the model with (one month abstinence aOR = 0.94, 95% CI 0.9–0.98, *p* = 0.009; six months abstinence aOR = 0.93, 95% CI 0.94–0.99, *p* = 0.014). Similarly, for quit attempts addition of these variables did not significantly alter the adjusted odds ratios for high TP (aOR = 1.18, 95% CI 1.01–1.39, *p* = 0.043). We also tested for non-linearity of the effect of DD by substituting continuous DD with categorical DD tertiles in our models predicting smoking abstinence. With low DD as the reference category, medium DD was not significant (aOR = 0.88, *p* = 0.169) and high DD (aOR = 0.71, *p* = 0.002) was highly significant for one month abstinence. Similarly, for six months abstinence, medium DD failed to reach significance (aOR = 0.91, *p* = 0.426) and high DD was highly significant with an even lower aOR than for one month abstinence (aOR = 0.60, *p* < 0.001). Taken together the direction and strength of the aORs do indicate linearity. Substituting household income (low, medium high) for financial distress in all models did not significantly alter the aORs for the main study variables.

Given the significant interactions for TP and DD by financial stress, the GEE models for smoking abstinence were stratified by financial stress (see Table 3). Among those smokers who reported financial stress there were no significant associations of time perspective or delay discounting with smoking abstinence. For those smokers who reported no financial stress, both lower time perspective and lower delay discounting were associated with successful abstinence maintenance.

## 4. Discussion

We found that TP and DD had distinctly different relationships both with making quit attempts and sustained abstinence from smoking consistent with other research showing they have different determinants [44,46] and our theoretical analysis, TP was predictive of making quit attempts, while DD was not. However, for predicting smoking abstinence, high DD predicted relapse, while TP did not, with a possible unexpected negative relationship with those disagreeing that they think about the future being most likely to succeed in the bivariate analysis, but only marginally in the multivariate analysis and only in those reporting no financial stress. Further, the measures of TP and DD were only very weakly related and positively, not negatively, contrary to our prediction.

The failure of DD to predict making quit attempts, even in bivariate analyses, suggests it is not influencing decisions to initiate quitting. These findings are contrary to expectations for theories which postulate that DD is assessing a generalized reduced valuing of the future, as quitting smoking is a clear case of prioritizing the future over the present and should shape future choices and thus influence intentions, albeit perhaps unconsciously. That there was also no association with either wanting to quit or plans to quit further emphasizes a lack of relationship with executive decisions that likely impact motivation to quit. The finding of a positive association between DD and increased risk of relapse is consistent with DD being considered an index of impulsiveness [2,8]. Low DD is thus an indicator of the capacity for restraint, capacities required to persist with the difficult task of maintaining abstinence from smoking, but arguably of little relevance for making choices.

Our findings also have implications for our understanding of TP. The results from this study confirm those of Hall et al. [20] which suggested that TP may influence attempts to quit but are unrelated to success in quitting at least when controlling for an increased tendency to try, a pattern also consistent with theoretical conceptualizations of time perspective [15]. As expected, those who agree they think more about the future (high TP) were more likely to make attempts to stop smoking and this occurred even when controlling for proximal smoking-specific predictors. However, we did not expect essentially no relationship between TP and DD nor the positive relationship between TP and relapse. Clearly, to plan for the future requires some tendency to think about future possibilities and the additive predictive power of TP on top of smoking-specific measures suggests that there is some characteristic of individuals that influences this over and above specific plans for smoking cessation. The remaining question is why the effects with remaining abstinent from smoking are different, indeed in some situations, negative.

The interactive effects with financial stress on quit success provides possible insights. For those not reporting financial stress at baseline, the relationships for both durations of abstinence were essentially the same. High DD and high TP (future focus) predicted relapse. By contrast, there were non-significant positive trends for high TP to predict success among those financially stressed for both cessation periods. It is also notable that those who were financially stressed reported higher levels of future thinking. Those who are in financial stress are facing situations in the present which threaten their future and working out what to do in the immediate future is practical and important and this may explain the increase in future thinking. However, whether this argument extends to simply thinking about any kind of future is unclear. We hypothesize that the increased focus on the future reported by those who are currently under financial stress is likely to be a focus on medium term issues around resolving the financial stress and/or not making the situation worse; that is, action to achieve goals over the near term; that is within weeks, or months. By contrast, thinking more generally about the future, particularly more distant futures without clear links to the current situation, may be more in the way of fantasizing [58]; that is, imagining futures with no linkages to mechanisms for their attainment, which is associated with a lower level of goal attainment. Alternatively, a focus on outcomes (e.g., health effects) or even goals without a commensurate consideration of mechanisms might have the same effects by not stimulating consideration of a path to the attainment of the goal. Thus, where future thinking is not practical such as failure to think through how future goals can be achieved, it would still motivate action, but leave the person unprepared to deal with the challenges associated with maintaining cessation. This would explain why the overall negative relationship between future orientation and success disappears in those reporting financial stress since the persons thinking in such cases is likely to be practical, and if it occurs in response to stressors, may be more likely to occur in other challenging contexts. To test these hypotheses will require separately assessing the extent to which future-oriented thoughts have a focus on potential achievability. It would be of interest to explore whether there is a stable measurable tendency of people to think practically about their future rather than in unrealistic, fantasizing ways.

In previous work where we have found motivational measures strongly predictive of smokers making quit attempts, we have interpreted a reverse effect on sustained cessation among those who try as potentially being due to these smokers being more dependent [59]. This analysis does not replace our hypothesis about those with high desire to quit and failing being more dependent, as can be the case among a population with a history of multiple failed attempts, rather it points to the need to focus on the extent to which smokers have thought about what is a practical way to achieve the desired goal if they are to overcome barriers to success. In this context, it would be of interest to see if the number and frequency of past quitting efforts was related to success on a given attempt.

What does it mean to value the future? The future as we conceive of it is really just a collection of ideas about possibilities that may happen or that we might be able to facilitate happening. To value the future means at the very least to have the ability to imagine futures in such a way that generate sufficient affective force to influence choices. Our analysis suggests that DD among daily smokers is partly driven by impulsivity, that is a tendency to act on actions strongly cued in the moment, rather than being able to inhibit these actions to allow for other considerations, of which predicted future consequences would be important. Clearly an ability to imagine future scenarios with some degree of credibility is going to be critical to inhibiting tendencies to act for immediate rewards, but this imaginative activity is only likely to affect behavior if it actually occurs at the time the undesirable impulses to act occur. The findings point to the importance of general dispositions that are not adequately represented in assessments of specific behaviours, such as DD, influencing operationally dominant activities such as sustaining smoking abstinence. Overall, findings are broadly as would be expected from CEOS theory and do not fit as well with a conceptualization of DD as measuring some fundamental undervaluing of the future. It would be useful for future studies to explore the relationships between DD and more direct measures of impulsivity and capacity to imagine scenarios which would seem to be its more basic determinants and to explore relationships with other difficult to change behaviours.

A recent large meta-analysis [16] concluded that a future time perspective had small-to-medium-sized positive associations with self-regulatory ability and specific aspects of this such as goal setting, goal operating (plans on how to achieve the goal), and also with outcomes. It is notable that none of their studies of future TP included the Hall and Fong measure [13,14] we took our item from, although several studies have explored its relationship to smoking outcomes at least. While our findings are consistent around goal setting as this is closely linked to quit attempts, the findings for quit success are not. They find positive associations with outcomes, we found a negative one, albeit conditional. However, a strength of our study is that our analysis of success was prospective and only among those trying, so is not confounded with factors that influence trying and thus the possibility of success. This is likely to be mainly an issue for difficult to sustain behaviours such as smoking cessation where self-regulation is required after acting as well as to act. As far as we can tell, none of the outcomes in the Baird et al. [16] corpus had smoking cessation, or other hard to maintain behaviours as an outcome or analyzed it among those making attempts. Thus, we don’t see this meta-analysis affecting our conclusions, but think our work highlights a limitation of treating future TP as an index of self-regulation: we agree it is likely to be an element supporting self-regulation as it applies to change, but not one that appears relevant to sustaining change.

Turning to the implications for smoking cessation, both TP and DD had predictive value over and above smoking specific predictors. However, the implications are likely to differ. In the case of DD, if indeed it is composed of ability to imagine futures when needed and impulsivity, we should be exploring the potential to increase people’s general skills in this area so they can apply them to complex tasks such as smoking cessation. It would seem that prior to actually having quit, smokers do not take into account the potential effect of their impulsivity on their likelihood of success, but this becomes manifest as they face the challenges of resisting temptations to smoke. That we found essentially the same effects for 6 months as for 1 months sustained abstinence, suggests some causal impacts of impulsivity occur in the early weeks of the quit attempt when ability to overcome the more immediate barriers to maintenance of cessation such as overcoming nicotine withdrawal are most likely to be evident. That said, we have not tested for differential effects of time quit by conducting analyses of relapse between 1 and 6 months to see if the importance of DD changes over time quit. In contrast, for TP, our findings suggests that we should be assessing the extent to which people think about their smoking, not just what they want but also about mechanisms for achieving their goals. That is, we should be encouraging smokers to focus on both what they should do and how they should do it if they are to increase their chances of turning desires to quit into successful cessation. Our findings also provide support for behaviour change interventions, that are designed to reduce the impacts of impulsiveness, such as trying to discourage major decision making at times when urges are strong, a strategy used in some effective smoking cessation interventions [60].

### Strengths and Limitations

Limitations. This study should not be used to estimate the magnitude of the predictive relationships because the measures were taken some time from the behaviours they are designed to predict (quit attempts and success) which would have attenuated these relationships. Further, some of the measures used are theorized to have large situational determinants (e.g., plans to quit), and all are subject to some situational variability, so we would expect much greater predictive utility if the gap between the measures and the outcomes was shorter. While both TP and DD are conceived as largely stable characteristics (more similar to traits than states), both constructs are potentially modifiable through situational factors. It would be interesting to assess them throughout a quit attempt to see if increases in DD or reductions in TP might predict relapse. In addition, for quit success, the predictor questions are asked while the person was smoking so the smokers had no current direct experience of the challenges of cessation, so a high level of prediction is never going to be likely.

Another major limitation is that for TP, we only used a single item from the TP scale developed by Fong and Hall [2003], so there is a higher degree of unreliability (although significant associations were obtained despite this). As we chose the “best” item in terms of item total correlation, our item is clearly in part measuring the core construct and, as noted, we see capacity to refine the measures to include consideration of pathways to future possibilities. Finally, longitudinal associations do not demonstrate causality, they still need to be argued for. We think causal mechanisms are plausible, but as should be clear from the above discussion, the plausibility of causation varies by what we think the measures are actually assessing. Given that changing DD has been shown experimentally to change choices, we think a causal role is plausible for its underlying mechanisms, at least to be affecting smoking cessation outcomes.

## 5. Conclusions

In conclusion, TP and DD are distinctly different concepts even though they both relate to effects of the future on choices and behaviours, in this case around smoking cessation. Our analyses support TP as being a measure of future thinking, but one that currently fails to differentiate possible futures from thinking about methods for pursuing them. In contrast, DD may be better thought of as indexing a capacity to evoke future possibilities, perhaps as a function of capacity to imagine them and to act in opposition to impulses to immediate gratification. That is, it is not so much about how often futures are considered but the ability to take them seriously when making choices that can have foreseeable future consequences.

## Figures and Tables

**Table 1 ijerph-19-00631-t001:** Delay discounting, time perspective, motivational, dependence and demographic characteristics by quit attempts and for >1 month and ≥6 months smoking abstinence among those smokers who made a quit attempt.

	Quit Attempts n = 6710			≥1 Month Smoking Abstinence n = 2743 *			≥6 Month Smoking Abstinence n = 2488 *		
Previous Wave Predictors	No Attempt n = 3900 %	Any Attempt n = 2810 %	*ES* *p*	<1 Month n = 1720 %	≥1 Month n = 1023 %	*ES* *p*	<6 Months n = 2044 %	>6 Months n = 444 %	*ES* *p*
Time perspective									
*Disagree*	28.9	18.5	0.16	16.6	21.9	0.07	16.5	22.1	0.06
*Neutral*	38.7	34.4	<0.001	34.1	34.0	0.001	34.7	32.4	0.019
*Agree*	32.3	47.1		49.3	44.1		48.8	45.5	
Delay Discounting M (SD)	4.79 (1.97)	4.88 (1.98)	0.04 0.072	4.98 (2.03)	4.70 (1.86)	0.14 <0.001	4.97 (2.03)	4.62 (1.74)	0.18 <0.001
Plan to quit									
*No*	47.3	18.6	0.39	16.6	21.3	0.06	16.9	22.7	0.09
*In future >6 months*	35.4	30.3	<0.001	30.3	30.9	0.010	29.5	33.8	<0.001
*Between 1–6 months*	13.8	34.2		35.8	31.7		36.2	27.0	
*Within 1 month*	3.5	16.9		17.3	16.1		17.3	16.4	
Want to quit									
*Not at all*	18.6	4.3	0.36	2.8	6.5	0.11	3.1	7.0	0.09
*A little*	23.6	11.3	<0.001	10.2	13.1	<0.001	10.5	12.4	<0.001
*Somewhat*	36.4	31.2		30.8	32.2		30.9	32.0	
*A lot*	21.4	53.2		56.3	48.2		55.5	48.6	
Perceived quit efficacy M (SD)	2.00 (1.14)	2.23 (1.12)	0.20 <0.001	2.19 (1.10)	2.28 (1.15)	0.09 0.029	2.20 (1.10)	2.29 (1.18)	0.07 0.16
Perceived addiction to smoking									
*None*	3.6	2.0	0.09	1.4	2.8	0.02	1.4	3.0	0.04
*Somewhat*	37.3	34.6	<0.001	32.5	37.7	<0.001	33.5	34.4	0.066
*Very*	59.1	63.4		66.1	59.6		65.1	62.6	
Strength of urge to smoke (0–5) M (SD)	2.74 (1.22)	2.85 (1.21)	0.09 <0.001	2.94 (1.18)	2.70 (1.23)	0.21 <0.001	2.91 (1.19)	2.75 (1.24)	0.13 0.014
HSI continuous (0–16) M (SD)	6.77 (2.14)	6.55 (2.08)	0.10 <0.001	6.68 (2.00)	6.35 (2.17)	0.15 <0.001	6.62 (2.03)	6.44 (2.19)	0.04 0.090
Age group (years)									
*18–24*	3.6	7.8	0.12	7.7	7.6	0.01	8.1	7.9	0.04
*25–39*	14.1	18.7	<0.001	19.1	18.7	0.982	19.0	16.2	0.370
*40–54*	34.1	31.1		31.2	31.0		31.2	30.2	
*≥55*	48.2	42.4		42.0	42.7		41.6	45.7	
Gender									
Male	48.6	43.9	0.05	44.4	43.7	0.01	44.4	46.4	0.02
Female	51.4	56.1	<0.001	55.6	56.3	0.711	55.6	53.6	0.450
Country									
*Canada*	25.6	33.4	0.13	35.6	30.2	0.01	35.7	28.6	0.07
*United States*	22.3	19.9	<0.001	8.7	20.9	0.003	18.5	21.8	0.007
*England*	35.5	25.3		23.4	28.2		23.7	29.1	
*Australia*	16.6	21.4		22.3	20.7		22.1	20.5	
Financial stress									
*Yes*	11.8	15.6	0.05	17.7	12.0	0.08	17.0	11.5	0.06
*No*	88.2	84.4	<0.001	82.3	88.0	<0.001	83.0	88.5	0.004
Education level attained									
*Low*	36.2	33.5	0.03	34.4	32.4	0.02	33.9	34.7	0.02
*Medium*	40.5	41.5	0.064	40.9	41.9	0.560	41.0	42.1	0.70
*High*	23.4	25.0		24.8	25.7		25.1	23.2	
Ethnicity									
*White*	90.7	89.0	0.03	88.7	89.4	0.01	88.3	91.4	0.04
*Non-White*	9.3	11.0	0.028	11.3	10.6	0.559	11.7	8.6	0.054

HSI: heaviness of smoking index; * after exclusion of n = 67 whose status was quit at follow-up but length of quit attempt was less than one month; associations are bivariate. ES: effect size (Cramer’s V for categorical data, Cohen’s d for continuous).

**Table 2 ijerph-19-00631-t002:** Generalized estimating equations (GEE) showing probability of quit attempts and for >1 month and ≥6 months quit success as function of delay discounting, time perspective, motivational, demographic and dependence measure variables.

	Quit Attempts n = 6579			>1 Month Smoking Abstinence n = 2705			>6 Months Smoking Abstinence n = 2452		
	aOR	95% Confidence Intervals	Sig	aOR	95% Confidence Intervals	Sig	aOR	95% Confidence Intervals	Sig
Time perspective									
*Disagree*	ref			ref			ref		
*Neutral*	1.16	[1.00, 1.35]	**0.044**	0.80	[0.64, 1.01]	0.065	0.73	[0.54, 0.99]	**0.044**
*Agree*	1.19	[1.02, 1.39]	**0.024**	0.80	[0.64, 1.00]	0.056	0.85	[0.64, 1.15]	0.295
Delay Discounting	1.01	[0.98, 1.04]	0.558	0.94	[0.90, 0.98]	**0.006**	0.93	[0.87, 0.98]	**0.008**
Plan to quit									
*No immediate plan*	ref			ref			ref		
*Beyond 6 months*	1.34	[1.15, 1.56]	**<0.001**	0.98	[0.76, 1.27]	0.886	0.99	[0.71, 1.36]	0.934
*Within 6 months*	2.81	[2.35, 3.37]	**<0.001**	0.88	[0.67, 1.16]	0.378	0.63	[0.44, 0.90]	**0.014**
*Within 1 month*	4.83	[3.78, 6.19]	**<0.001**	0.91	[0.66, 1.25]	0.574	0.75	[0.50, 1.13]	0.173
Want to quit									
*No desire*	ref			ref			ref		
*A little*	1.62	[1.27, 2.07]	**<0.001**	0.56	[0.35, 0.90]	**0.016**	0.60	[0.34, 1.07]	0.081
*Somewhat*	2.21	[1.74, 2.80]	**<0.001**	0.47	[0.30, 0.74]	**0.001**	0.58	[0.34, 1.00]	0.051
*A lot*	4.02	[3.12, 5.18]	**<0.001**	0.42	[0.26, 0.66]	**<0.001**	0.54	[0.31, 0.95]	**0.031**
Perceived quit efficacy	0.98	[0.92, 1.03]	0.390	1.09	[1.01, 1.18]	**0.030**	1.14	[1.03, 1.27]	**0.014**
HSI (scale 0–16)	0.97	[0.94, 1.00]	0.090	0.95	[0.91, 1.00]	0.062	0.96	[0.90, 1.03]	0.287
Strength of urges to smoke	1.04	[0.98, 1.10]	0.164	0.91	[0.84, 0.99]	**0.034**	0.94	[0.84, 1.05]	0.282
Perceived addiction to smoking									
*Not at all*	ref			ref			ref		
*Somewhat addicted*	1.04	[0.72, 1.51]	0.820	0.87	[0.47, 1.58]	0.637	0.74	[0.36, 1.53]	0.434
*Very addicted*	1.06	[0.72, 1.57]	0.748	0.92	[0.49, 1.71]	0.783	0.88	[0.41, 1.90]	0.761
Age Group									
*18–24*	2.08	[1.60, 2.69]	**<0.001**	0.96	[0.68, 1.34]	0.804	0.94	[0.60, 1.45]	0.772
*25–39*	1.11	[0.94, 1.32]	0.207	1.08	[0.85, 1.37]	0.534	0.91	[0.66, 1.25]	0.545
*40–54*	0.84	[0.74, 0.96]	**0.011**	1.09	[0.89, 1.32]	0.411	0.98	[0.76, 1.26]	0.870
*Over 55*	ref			ref			ref		
Gender									
*Male*	ref			ref			ref		
*Female*	1.09	[0.97, 1.22]	0.145	1.06	[0.90, 1.25]	0.513	0.94	[0.76, 1.17]	0.594
Country									
*England*	ref			ref			ref		
*Canada*	1.32	[1.14, 1.54]	**<0.001**	0.76	[0.61, 0.95]	**0.017**	0.68	[0.51, 0.90]	**0.008**
*United States*	1.12	[0.95, 1.33]	0.174	0.99	[0.77, 1.28]	0.955	0.98	[0.71, 1.36]	0.924
*Australia*	1.51	[1.27, 1.79]	**<0.001**	0.86	[0.67, 1.09]	0.210	0.80	[0.58, 1.09]	0.161
Financial stress									
*No*	ref			ref			ref		
*Yes*	1.01	[0.86, 1.19]	0.888	0.75	[0.59, 0.96]	**0.019**	0.80	[0.57, 1.11]	0.176
*Tests for interactions*									
TP × DD	1.02	[0.98, 1.06]	0.274	1.02	[0.97, 1.08]	0.407	1.00	[0.93, 1.08]	0.996
TP × Country									
England	ref			ref			ref		
Canada	1.02	[0.66, 1.57]	0.227	0.82	[0.62, 1.09]	0.174	0.86	[0.59, 1.26]	0.436
United States	1.09	[0.69, 1.71]	0.860	0.93	[0.68, 1.28]	0.656	0.99	[0.66, 1.51]	0.982
Australia	1.59	[0.99, 2.56]	0.837	1.06	[0.78, 1.43]	0.700	1.13	[0.75, 1.69]	0.564
TP × Age Group	1.07	[0.90, 1.28]	0.441	0.86	[0.67, 1.10]	0.241	0.95	[0.68, 1.33]	0.764
TP × Financial Stress	1.03	[0.83, 1.27]	0.811	0.72	[0.51, 0.99]	0.050	0.56	[0.34, 0.92]	**0.029**
TP × Education	1.04	[0.95, 1.14]	0.374	1.05	[0.92, 1.21]	0.483	0.92	[0.76, 1.11]	0.379
TP × Ethnicity	1.25	[0.99, 1.57]	0.056	1.18	[0.82, 1.72]	0.372	1.01	[0.59, 1.73]	0.973
TP × HSI	1.03	[0.99, 1.06]	0.145	0.98	[0.93, 1.03]	0.374	0.96	[0.90, 1.04]	0.328
DD × Country									
England	ref			ref			ref		
Canada	0.93	[0.86, 1.01]	0.076	1.08	[0.96, 1.21]	0.206	1.03	[0.89, 1.20]	0.665
United States	1.02	[0.94, 1.10]	0.700	1.05	[0.92, 1.19]	0.457	1.06	[0.90, 1.23]	0.498
Australia	1.00	[0.92, 1.09]	0.966	0.95	[0.84, 1.08]	0.438	0.96	[0.82, 1.12]	0.612
DD × Age Group	1.02	[0.96, 1.09]	0.510	0.94	[0.85, 1.03]	0.163	0.97	[0.85, 1.10]	0.598
DD × Financial Stress	1.03	[0.96, 1.12]	0.457	0.88	[0.79, 0.99]	**0.027**	1.04	[0.88, 1.23]	0.567
DD × Education	1.00	[0.96, 1.04]	0.872	1.04	[0.98, 1.09]	0.201	1.00	[0.93, 1.08]	0.929
DD × Ethnicity	1.01	[0.93, 1.10]	0.831	1.05	[0.93, 1.19]	0.389	1.10	[0.93, 1.31]	0.250
DD × HSI	0.99	[0.98, 1.00]	0.169	0.99	[0.97, 1.01]	0.293	0.99	[0.97, 1.02]	0.713

Other covariates in the full model include ethnicity, education level attained, smoking permitted in the home (allowed, sometimes, never), and non-daily vaping frequency (weekly to monthly, non-vaper). All tests for interactions conduced individually within fully adjusted models. aOR adjusted odds ratio. Bolded figures statistically significant at at least *p* < 0.05.

**Table 3 ijerph-19-00631-t003:** Generalized estimating equations (GEE) showing probability of >1 month and ≥6 months quit success as function of delay discounting, time perspective, motivational, demographic and dependence measure variables: stratified by financial distress.

	>1 Month Smoking Abstinence						>6 Months Smoking Abstinence					
	Financially Distressed n = 422			Not Financially Distressed n = 2283			Financially Distressed n = 393			Not Financially Distressed n = 2059		
	aOR	95% CI	Sig	aOR	95% CI	Sig	aOR	95% CI	Sig	aOR	95% CI	Sig
Time perspective												
*Disagree*	ref			ref			ref			ref		
*Neutral*	0.62	[0.28, 1.34]	0.209	0.82	[0.64, 1.05]	0.121	0.96	[0.27, 3.43]	0.943	0.72	[0.52, 0.98]	**0.037**
*Agree*	1.18	[0.59, 2.38]	0.634	0.75	[0.59, 0.96]	**0.024**	2.24	[0.71, 7.11]	0.163	0.77	[0.57, 1.05	0.105
Delay discounting	1.05	[0.94, 1.18]	0.342	0.92	[0.88, 0.97]	**0.001**	0.90	[0.76, 1.07]	0.143	0.93	[0.88, 0.99]	**0.032**
Plan to quit												
*No immediate plan*	ref			ref			ref			ref		
*Beyond 6 months*	1.22	[0.58, 2.60]	0.615	0.95	[0.72, 1.25]	0.724	1.44	[0.50, 4.11]	0.497	0.97	[0.69, 1.37]	0.862
*Within 6 months*	0.57	[0.26, 1.26]	0.176	0.94	[0.70, 1.26]	0.665	0.57	[0.19, 1.77]	0.331	0.65	[0.44, 0.96]	**0.032**
*Within 1 month*	0.57	[0.22, 1.46]	0.253	0.95	[0.68, 1.34]	0.792	0.63	[0.17, 2.41]	0.503	0.77	[0.50, 1.20]	0.256
Want to quit												
*No desire*	ref			ref			ref			ref		
*A little*	0.34	[0.08, 1.35]	0.130	0.56	[0.34, 0.92]	**0.023**	0.70	[0.09, 5.15]	0.761	0.58	[0.32, 1.05]	0.070
*Somewhat*	0.55	[0.15, 2.00]	0.376	0.44	[0.27, 0.71]	**0.001**	0.60	[0.09, 4.11]	0.654	0.56	[0.32, 0.99]	**0.040**
*A lot*	0.48	[0.13, 1.75]	0.282	0.39	[0.24, 0.64]	**<0.001**	0.61	[0.09, 4.13]	0.668	0.52	[0.29, 0.94]	**0.027**
Perceived quit efficacy	1.17	[0.92, 1.49]	0.198	1.09	[1.00, 1.19]	0.052	1.20	[0.86, 1.67]	0.295	1.13	[1.01, 1.27]	**0.027**
HSI (scale 0–16)	1.07	[0.93, 1.23]	0.341	0.94	[0.89, 0.99]	**0.033**	1.12	[0.92, 1.37]	0.225	0.95	[0.89, 1.02]	0.167
Strength of urges to smoke	0.90	[0.72, 1.13]	0.375	0.91	[0.83, 1.00]	**0.049**	0.81	[0.59, 1.12]	0.158	0.97	[0.86, 1.09]	0.564
Perceived addiction to smoking												
*Not at all*	ref			ref			ref			ref		
*Somewhat addicted*	0.63	[0.11, 3.49]	0.588	0.92	[0.48, 1.75]	0.789	0.90	[0.08, 9.71]	0.920	0.74	[0.34, 1.60]	0.471
*Very addicted*	0.31	[0.05, 1.81]	0.184	1.08	[0.55, 2.12]	0.816	0.71	[0.06, 8.09]	0.742	0.94	[0.42, 2.12]	0.891
Age Group												
*18–24*	1.30	[0.56, 3.05]	0.545	0.96	[0.66, 1.40]	0.837	1.11	[0.34, 3.61]	0.857	0.93	[0.57, 1.52]	0.782
*25–39*	1.48	[0.76, 2.86]	0.255	1.09	[0.84, 1.41]	0.532	1.34	[0.54, 3.31]	0.522	0.87	[0.61, 1.23]	0.440
*40–54*	1.81	[0.99, 3.31]	0.060	1.05	[0.85, 1.29]	0.668	1.30	[0.56, 3.03]	0.553	0.97	[0.74, 1.27]	0.805
*Over 55*	ref			ref			ref			ref		
Gender												
*Male*	ref			ref			ref			ref		
*Female*	1.10	[0.67, 1.81]	0.704	1.05	[0.88, 1.26]	0.572	0.99	[0.49, 1.97]	0.967	0.94	[0.74, 1.18]	0.575
Country												
*England*	ref			ref			ref			ref		
*Canada*	1.14	[0.56, 2.34]	0.724	0.73	[0.57, 0.92]	0.009	0.67	[0.25, 1.80]	0.419	0.67	[0.50, 0.91]	**0.011**
*United States*	1.63	[0.75, 3.56]	0.218	0.93	[0.71, 1.22]	0.597	1.54	[0.55, 4.35]	0.420	0.90	[0.64, 1.28]	0.562
*Australia*	1.08	[0.49, 2.39]	0.853	0.85	[0.66, 1.10]	0.220	0.88	[0.29, 2.66]	0.828	0.79	[0.57, 1.11]	0.173

aOR adjusted odds ratio, CI confidence interval. Other covariates in the full model include ethnicity, education level attained, smoking permitted in the home (allowed, sometimes, never), and non-daily vaping frequency (weekly to monthly, non-vaper).

## Data Availability

The data are jointly owned by a third party in each country that collaborates with the International Tobacco Control Policy Evaluation (ITC) Project. Data from the ITC Project are available to approved researchers 2 years after the date of issuance of cleaned data sets by the ITC Data Management Centre. Researchers interested in using ITC data are required to apply for approval by submitting an International Tobacco Control Data Repository (ITCDR) request application and subsequently to sign an ITCDR Data Usage Agreement. To avoid any real, potential, or perceived conflict of interest between researchers using ITC data and tobacco-related entities, no ITCDR data will be provided directly or indirectly to any researcher, institution, or consultant that is in current receipt of any grant monies or in-kind contribution from any tobacco manufacturer, distributor, or other tobacco-related entity. The criteria for data usage approval and the contents of the data usage agreement are described online (http://www.itcproject.org). The authors of this paper obtained the data following this procedure. This is to confirm that others would be able to access these data in the same manner as the authors. The authors did not have any special access privileges that others would not have.

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
