# Peer review of "The Predictive Utility of Valuing the Future for Smoking Cessation: Findings from the ITC 4 Country Surveys"

_ijerph, 2022, doi:10.3390/ijerph19020631_

Round 1
Reviewer 1 Report
I found rigorous and correct the methodology of selection and evaluation of data here presented. The authors investigate whether the relationship between TP and DD for quitting and sustaining cigarette abstinence is moderated by socio-demographic factors, financial stress, addiction and motivation measures.
However, they should also consider the effect of their health state on smoking cessation. In fact, it is known that in most cases the choice to stop smoking depends on the manifestation of clinical events for which smoking is a risk factor.
Author Response
The reviewer is correct, health status is an important factor influencing quitting decisons, but typically moderated through quit motivations, which we do control for. The meaures we had on perceived health were also included, but are of less importance when more proximal determinants are included in multivariate analysis. In population studies few have newly diagnosed, smoking-related health issues, so it is not practical to explore such issues (this is for clinical studies).
Reviewer 2 Report
Dear colleagues, your manuscript has an exciting line of research, and it is indeed innovative. However, it needs substantial work to be published.
- The introduction is long and confusing. There is the need to present your background and significance in a way that justifies the selection of DD and TP, especially when they are not interrelated.
- In the methods, it is unclear how you choose the questions used as proxies for the constructs among the items available in the inventories. This part must show other authors' work on using those questions as valid proxies.
- The results and discussion need more alignment with what you proposed as the hypotheses, methods, and measures.
Author Response
We accept the paper is complex, as is inevitable when dealing with complex concepts that many people see as equivalent, but which we thoerised and demonstrated to be quite different, although related. The concerns about complexity are adressed in response to Reviewer 3, who provided some specific suggestions ias to how to improve readability.
Reviewer 3 Report
This is an interesting paper based on analysis of a large, multi-country cohort of smokers (ITC study). The paper seeks to assess the predictive value of two related but conceptually distinct measures of valuation of future outcomes relevant to smoking cessation: quit attempts and sustained abstinence over 1 and 6 months. Time Perspective (TP) is conceived as a belief about future intentions, rather than a specific behavioral outcome. In contrast, Delay Discounting (DD) is conceived as a direct behavioral measure of competing valuations of earlier (or present) versus later (or future) reward. The authors use the unique opportunity presented with the availability of successive survey waves in their multi-country study to assess the relationship between TP and DD, and the predictive capacity for each construct on cessation outcomes. Based on the characterization of TP as an index of future intentions (but not behavioral goal attainment), the authors hypothesize that TP will predict quit attempts but not sustained abstinence. Likewise, because DD is characterized as an index of behavioral preference for earlier reward (and thus impulsiveness), it is hypothesized as a predictor of sustained abstinence but not future quit attempts. Theory-guided proposed moderators of these effects were also explored, including financial instability and smoking dependence measures.
The methods and analyses are rigorous, yielding results that are largely as hypothesized. The findings have the potential provide valuable new insights into mechanisms of appraisal of future smoking cessation goals. Because these mechanisms were assessed in parallel, and were found to influence different cessation outcomes, these findings may provide a basis for more finely tailored cessation intervention strategies.
Despite the overall promise of this paper, there are a number of critical steps the authors could take to strengthen communication of the central premise and communicate the findings more effectively.
Chiefly, the outline of the conceptual basis for the study objectives in the introduction is remarkably difficult to navigate. The theoretical overview seems excessively long-winded, characterized by unnecessary detail and challenging sentence construction. A prime example of this problem includes content related to CEOS theory, covering passages from lines 89 – 171. While CEOS theory is of interest, the long and complicated outline does not substantially advance the reader’s capacity to appreciate the central premise of the study. For example, considerable emphasis is placed on the role of “stories” which is not critical to explain the basic research premise. Perhaps more telling is the fact that “stories” are not evoked in the discussion to explain of contextualize the findings. The authors are strongly urged to reduce and simplify this content and allow the reader to focus on understanding the major knowledge gap and the resulting research objectives.
A related problem is a complicated and sometimes undisciplined style of sentence construction, which makes key arguments unnecessarily hard to follow. For example, the sentence at lines 81-83 makes a critical point, but is convoluted: “beliefs about dispositions” is hard to reconcile when beliefs are explained as distinct from dispositions in the next paragraph (line 89). Other examples might include sentences at lines 55-59 and 68-72 (this latter one also being repetitious).
Lastly, while the discussion is considerably more direct than the introduction, there is a notable lack of consideration on how these findings might be positioned to inform the refinement of cessation interventions. Some additional insight on this question would be helpful.
Minor issues:
Some interesting findings are referred to in Appendix A and Supplementary Table 1, yet these tables are not attached to the manuscript or available elsewhere on the manuscript submission site. What is the reason for the different descriptions (i.e. an appendix vs. a supplementary table)? Would it be possible to fold these tables into a single one and present them as part of the main paper?
There is a considerable number of basic errors/oversights in the written content. Some noted examples are below. The authors are encouraged to undertake a more through proof reading to ensure the best possible presentation of their otherwise excellent paper.
Line 31: “was negative associated with” (should be “negatively”)
Line 71/72: explain the time-frame implied by “longer periods”.
Line 76: an example of several instances of a missing article “the” or “a” (“about extent to which”; also, “on implications of”). Also see lines 175, 177, 200/201, 203.
Line 142: “bodies” should be “body’s”
Line 143: not clear why nicotine is followed by “(X)”. Suggest delete.
Line 148: what is the “affective force for not smoking”?
Line 150 “overtime” should be “over time”
Line 151: “smokers” needs an apostrophe: “smoker’s”
Line 596: Typo: “they still needs to be argued for”
Line 604: “Our analyses supports”
Citations required:
Line 64/65: Last part of the sentence to support “reduced likelihood of relapse”.
Line 139-141
Line 456
Author Response
Overall, the Reviewers commented on the complexity. We have strugglied with this. Some of the concepts we are exploring are not well described or differentated in colloquial english, and as we are trying to write for a broad audience , we have tried to make key distinctions clear. We re-read and have made some changes to improve readability, but recognise it is still hard going for someone for whom these concepts are novel.
The specific suggestions made were valuable in this regard. Maintaining a copmmon voice in a paper with many authors is a challenge , and we clearly failed in places, but think the revision is now a more coherent read. We also reproofed read and corrected typos we found and added references as requested.
In particular , we have reviewed the section on CEOS theory and shortened it somewhat. We still keep the mterial on sories as it is central to the argument, but agree it is not raised in the discussion as its relevance is to an understanding of why DD and TP differ, not how, which is the focus of the results and thus discussion.
Round 2
Reviewer 1 Report
I approve the manuscript for publication.